# Molecular Dynamics Study of Interfacial Micromechanical Behaviors of 6H-SiC/Al Composites under Uniaxial Tensile Deformation

**DOI:** 10.3390/nano13030404

**Published:** 2023-01-19

**Authors:** Kai Feng, Jiefang Wang, Shiming Hao, Jingpei Xie

**Affiliations:** 1School of Physics and Microelectronics, Zhengzhou University, Zhengzhou 450052, China; 2School of Physics and Engineering, Henan University of Science and Technology, Luoyang 471023, China; 3School of Materials Science and Engineering, Henan University of Science and Technology, Luoyang 471023, China

**Keywords:** molecular dynamics, interface, tensile, dislocation movement, deformation mechanism

## Abstract

This paper investigated the micromechanical behavior of different 6H-SiC/Al systems during the uniaxial tensile loading by using molecular dynamics simulations. The results showed that the interface models responded diversely to the tensile stress when the four low-index surfaces of the Al were used as the variables of the joint surfaces. In terms of their stress–strain properties, the SiC(0001)/Al(001) models exhibited the highest tensile strength and the smallest elongation, while the other models produced certain deformations to relieve the excessive strain, thus increasing the elongation. The SiC(0001)/Al(110) models exhibited the largest elongations among all the models. From the aspect of their deformation characteristics, the SiC(0001)/Al(001) model performed almost no plastic deformation and dislocations during the tensile process. The deformation of the SiC(0001)/Al(110) model was dominated by the slip of the 1/6 <112> Shockley partial dislocations, which contributed to the intersecting stacking faults in the model. The SiC(0001)/Al(111) model produced a large number of dislocations under the tensile loading. Dislocation entanglement was also found in the model. Meanwhile, a unique defect structure consisting of three 1/6 <110> stair-rod dislocations and three stacking faults were found in the model. The plastic deformation in the SiC(0001)/Al(112) interface model was restricted by the L-C lock and was carried out along the 1/6 <110> stair-rod dislocations’ direction. These results reveal the interfacial micromechanical behaviors of the 6H-SiC/Al composites and demonstrate the complexity of the deformation systems of the interfaces under stress.

## 1. Introduction

As a kind of composites material, SiC-reinforced aluminum composites (SiC/Al) have already been extensively applied in diverse fields, such as aerospace, automotive, electronic packaging, etc., due to their high specific strength, low thermal expansion, and many other physical properties [1,2,3]. The interface of composite materials is the area with significant changes in the chemical composition between the matrix and reinforcement, which can combine with each other and play a role in the load transfer [4,5]. In recent years, growing interests in high-volume SiC/Al composites and SiC/Al nanocomposites [6,7,8,9,10,11] have raised the importance of the interfaces in SiC/Al composites, for the ratio of the interface in the composites rises when the size of the reinforcement material decreases, or the volume of the reinforcement material increases. Therefore, research is necessary to carry on the interfaces of SiC/Al composites.

Experimental methods were applied to study the microstructure and mechanical properties of SiC/Al interfaces. Guo et al. [12] conducted uniaxial compression on micro-pillars with the SiC/Al system. They explained the observation results by the grain fragmentation and dislocation pile-up at the SiC/Al interface upon deformation. Their study highlighted the importance of interfaces in the deformation mechanism of particle-reinforced metal matrix composites. Gong et al. [13] conducted an in situ TEM (Transmission Electron Microscope) tensile test on SiC/Al with stacking faults (SFs) and observed the interaction between SFs and dislocation arrays. Their study revealed how SFs influence the plastic deformation in SiC/Al composites. Although studying ceramic/metal interfaces through experiments is direct and reliable, great difficulties appear regarding the preparation and observation of samples and the complex interfacial microstructure of composite materials.

With the rapid development of computational materials science, simulating calculation methods at the atomic level has access to studying the complex ceramic/metal interface micro-regions. By using the first principal calculation method, Wu et al. [14,15] studied the influence of the introduction of point defects on the bonding of the 6H-SiC(0001)/Al(111) interface. Liu et al. [16] studied the effect of Mg atoms on the adhesion properties of the 3C-SiC(111)/Al(111) interface, and their results showed that the Mg-element’s addition could significantly improve the covalent bond strength of Si-Al and C-Al, thereby improving the interfacial strength. By using the molecular dynamics (MD) method, Luo et al. [17] calculated the cohesive energy of 15 different SiC/Al interfacial systems. In their study, the correlations between the interface of specific binding relationships and their cohesive internal energy were determined. Huo et al. [18] conducted a molecular dynamics simulation to study the micro deformation mechanism of SiC/Al nanocomposites under tensile stress. Their results showed that the addition of SiC particles could significantly improve the tensile strength of the composites and demonstrated that the strengthening mechanism of the SiC/Al composites mainly included thermal mismatch strengthening, Orowan strengthening, and a loading transfer effect. Although many MD simulations were carried out to investigate the interfaces of the SiC/Al composites, there is little research involving how the dislocation movement and atomic deformation respond to the strain.

The aim of this paper, therefore, is to study how different interface systems of SiC/Al composites respond to tensile loading perpendicular to the interface. The simulation method chosen was the MD method. For the MD simulation, the energy, velocity, and force of atoms can be solved by the interaction between the atoms described by Newton’s equation and the empirical force field, which makes the simulation of thousands to millions of atoms feasible. The low-index surfaces of a crystal structure have a relatively low surface energy, which makes them easier to appear in the experiment. Therefore, the Al surfaces chosen in the simulations were the (001), (110), (111), and (112) planes. The other side of the interface was the 6H-SiC(0001) planes with Si terminated and with C terminated. In this study, therefore, uniaxial tensile tests were conducted on eight different 6H-SiC/Al interface models through the molecular dynamics method. The deformation of the interface was monitored to obtain the main mechanical property parameters and to explore the dynamic process of atomic migration, stress–strain characteristics, and dislocation interactions. The internal mechanism of the 6H-SiC/Al interface microzone from the perspective of the atomic motion and the dislocation movement was investigated.

## 2. Computational Models and Methods

### 2.1. Details of MD

This study used the open-source program LAMMPS to conduct the MD simulation [19]. Every model system was composed of a double-layer structure of 6H-SiC and Al, as SiC(0001)-Si/Al(001) interface model shown in Figure 1, with their respective crystal structures [20,21] listed in Table 1. Slight adjustments were carried out to the sizes of the boundaries in every model to ensure the free boundaries’ periodicity, as details are shown in Table 2. Fixed layers were set at the upper and lower boundary parts of the model to avoid the influence of the periodic boundary conditions.

A uniform deformation can help avoid shock waves generated in the simulation [22,23]. Therefore, during MD simulations of uniaxial tensile deformation, the system was uniformly stretched in the Z-direction at each time step. The time step set in the simulation was 0.001 ps. Before adding the tensile loading, the model was relaxed at a temperature of 300 K for 50 ps to eliminate the internal stress in the model. The strain rate of the tensile process is 5 × 10^10^ s^−1^ and then ran the simulation for 60,000 steps to reach the target shape variable, i.e., ε = 0.3. The engineering strain ɛ was defined by
(1)$$\varepsilon =\frac{L-L_{0}}{L_{0}}$$
where L is the instantaneous length on the *z*-axis of the model, and L_0_ is the initial length on the *z*-axis of the model. The whole stretching process was completed under the condition of an isothermal–isobaric (NPT) ensemble, with the Nose–Hoover thermostat to ensure temperature stability, and the pressure in the *x*-axis and *y*-axis directions was kept at zero during the stretching process. In this study, the Virial contributions of atoms were divided by the entire cell volume to calculate the stress of the entire system, known as Virial expression [24].

Another open-source software OVITO (version: ovito.3.7.2), was used to visualize atomic structure evolution [25,26]. The polyhedral template-matching method (PTM) helped complete the microstructure analysis results. This method can identify structures such as FCC, BCC, HCP, etc. The discontinuity extraction algorithm (DXA) was applied to evaluate the distribution of dislocation lines during stretching. The DXA converts the original atomic representation of a dislocated crystal into a line-based representation of a dislocation network and determines the Burgers vector for each dislocation. The algorithm can identify partial dislocations and some secondary grain boundary dislocations [27].

### 2.2. Potential

Different from the ab initio molecular dynamics simulation [28,29], the classical molecular dynamics simulation describes atomic motion based on the empiric force field adopted for the interaction between atomic nuclei, ignoring the effect of electrons. Therefore, for MD simulation, a reliable force field is vital to reasonable results. In this study, the mixed force field was adopted to describe the atomic interaction. The EAM (Embedded Atom Method) model developed by Mishin et al. [30] was used to simulate bulk aluminum. The Tersoff potential [31] was used to describe the interactions between Si and C atoms. For the interaction of C-Al and Si-Al, Dandekar et al. [32] obtained Morse potential function parameters based on data from Zhao et al. [33] and proved its accuracy with the paired potential curves of C-Al and Si-Al. Therefore, the SiC/Al interfaces in this study were described by Morse potential with the parameters summarized in Table 3.

## 3. Results and Discussion

### 3.1. Analysis of Stress–Strain Characteristics of the SiC(0001)/Al Interface Models

The engineering stress–strain curves of eight models under tensile loading are shown in Figure 2, along with the detailed figure of their elongations, tensile strength, and Young’s modulus, as shown in Table 4. It can be seen from Figure 2 that changes in the terminal atoms type of the SiC(0001) plane had little influence on the regularity of the stress–strain curves, and there were only a few differences that existed in the elongations and tensile strength of each pair. However, the stress–strain curves exhibited great diversity when changing the Al plane of the interfaces. The SiC(0001)/Al(001) models broke with the highest tensile strength and smallest elongation, and the relationships between their stress and strain maintained linearity before fracture, meaning that the SiC(0001)/Al(001) models suffered higher stress under the condition of less strain. The other models, except for the SiC(0001)/Al(001) models, exhibited larger elongations with visible yielding phenomena, but their tensile strength was relatively low. It was evident from Table 4 that the elongations of the SiC(0001)/Al(110) models were the largest of all, and the models with a higher Young’s modulus had lower elongations.

### 3.2. Wigner–Seitz Defect Analysis on the SiC(0001)/Al Interface Models

A Wigner–Seitz defect analysis was conducted on eight SiC(0001)-Si/Al interface systems of relaxed states. The Al-atoms which were not in the original lattice space were colored yellow, as shown in Figure 3. It was noted that the Al-atoms in the interface region of eight models deviated their original FCC lattice spaces at different amounts, which was driven by the stronger Si-Al bonding and C-Al bonding compared to the Al-Al bonding [34]. Therefore, the Al-atoms two to three layers away from the heterogeneous interface were affected due to a lattice mismatch, and their normal placing mode was also disturbed. However, it can be seen from Figure 3c,g that most Al-atoms in the SiC(0001)/Al(111) model were located on the standard lattice space, indicating that the SiC(0001)/Al(111) model showed the least lattice mismatch.

Considering that the switches of the Si or C terminating modes of the SiC(0001) plane had little influence on the consequences, studies on the atomic deformation and the dislocations of the interface models composed of SiC(0001)-Si and four Al low-index planes were conducted in the following studies. 

### 3.3. Analysis of Tensile Processes of Four SiC(0001)-Si/Al Models

Figure 4 shows the snapshots of the atomic strain tensor of the four SiC(0001)-Si/Al interface models during the tensile loading. Moreover, the stress of all the atoms in the region from x = 15 to x = 20 was calculated to obtain the stress distribution of the four SiC(0001)-Si/Al interface models, as shown in Figure 5. The four different interface models exhibited a distinct diversity in the atomic deformation under tensile loading, as can be seen in Figure 4. Meanwhile, it is noted that there was almost no deformation in the SiC matrix in all the interface models. The atomic strain in the SiC(0001)-Si/Al(001) model was uniformly distributed through the whole deformation process, before it fractured at 0.075 (Figure 4a,b). It was observed from Figure 5a that the stress concentrated on the interface region, where the crack started to nucleate and grow (Figure 4c,d). (As a matter of fact, the interface region mentioned here means the specific area two or three atomic layers away from the Si-Al heterogenous bonding interface in the Al matrix [34,35,36]. This fracture mode may be attributed to the unique mechanisms of the interface combination mode as mentioned above). For the deformation in the SiC(0001)-Si/Al(110) model, several intersecting stacking faults (Sfs) were detected after the elastic stage, as shown in Figure 4e,f. As the strain continued to rise and reached 0.136, the atom distance in the Al matrix was dramatically expanded. Although the stress distribution in the Al matrix was relatively uniform, there was still stress concentrated at the interface region (Figure 5b). Subsequently, the crack nucleated at the interface region because of the stress concentration (Figure 4g,h). A wavy deformation zone was discovered in the Al matrix of the SiC(0001)-Si/Al(111) model at the elastic stage, as shown in Figure 4i,j. Some deformation did not completely penetrate the Al matrix and reach the rigid layer (fixed region). When the strain increased to 0.123, the stress distribution was quite uneven, and the stress was distributed at the place where the atomic deformation was large, as can be seen from the comparison between Figure 4k and Figure 5c. Different from other models, the cracks of this model nucleated not only at the interface region but also in the Al matrix (Figure 4l). This phenomenon can be explained by the promising compatibility between the SiC(0001)-Si plane and Al(111) plane, as discussed in Section 3.2. As for the SiC(0001)-Si/Al(112) model, the same as the others, it experienced an elastic stage at the beginning of the loading (Figure 4m). Afterward, deformation zones extending in the [110] direction were found in the model (Figure 4n). With the further increase in the strain, the stress concentrated in the deformation zones and the interface region, as can be seen in Figure 4o and Figure 5d. Finally, the crack of the SiC(0001)-Si/Al(112) model appeared at the interface region (Figure 4p).

### 3.4. Structure and Dislocation Analysis of Four SiC(0001)-Si/Al Models

In order to better understand the deformation behaviors observed in Figure 4, the atomic structure and dislocation of four SiC(0001)-Si/Al interface systems were studied during the tensile process. Figure 6 illustrates the variation in the dislocation density of the four models under tensile loading. Moreover, Figure 7 shows snapshots of the typical structures and the corresponding dislocation at each specific strain of the four interface models. It is evidenced from Figure 6 that there was almost no generation of a dislocation during the tensile loading on the SiC(0001)-Si/Al(001) model. The FCC structure was also maintained through the whole deformation process, as can be seen in Figure 7a,b, which conforms to the results observed in Figure 3a–d. The dislocations in the SiC(0001)-Si/Al(110) model began to generate when the strain reached 0.035. In the meantime, the 1/6 <112> Shockley partial dislocations nucleated in the Al matrix and arched out from the interface region. Then, the dislocations glided along the [110] direction on the {111} plane of the Al matrix, leaving intersecting stacking faults; the corresponding snapshots are shown in Figure 7c,d. The dislocation continuously moved and reacted until the strain reached 0.11, when the number of the dislocations detected in the model began to show a downward trend, as can be seen in Figure 6. The reason for this phenomenon can be attributed to the large elongations of the SiC(0001)-Si/Al(110) model. At the later deformation stage, the model is excessively stretched, which makes the Al-atoms arranged irregularly, and the dislocation is no longer detected. As for the SiC(0001)-Si/Al(111) model, the dislocation density ascended rapidly when the strain increased to 0.049, and multiple 1/6 <112> Shockley partial dislocations were activated simultaneously. Some dislocations reacted during the movement to generate the 1/6 <110> stair-rod dislocations, contributing to a unique dislocation configuration in the Al matrix (Figure 7i), which consisted of three 1/6 <110> stair-rod dislocations lines and three stacking faults. As the strain continued to rise, a large number of dislocations expanded and entangled with each other, forming a massive deformation in the Al matrix (Figure 7e,f). Finally, there was a slight downward trend in the dislocation density as the SiC(0001)-Si/Al(110) model did. The dislocation in the SiC(0001)-Si/Al(112) model occurred at a strain of 0.046. In addition, the dislocation reaction contributed to a stable dislocation configuration (Figure 7j), which consists of two 1/6 <112> Shockley partial dislocation lines, one 1/6 <110> stair-rod dislocation line, and two stacking faults, also known as the Lomer–Cottrell (L-C) locks [37]. This dislocation configuration restricted the continuation of the deformation in certain directions due to the inoperable nature of the 1/6 <110> stair-rod dislocations. Hence, the 1/6 <112> Shockley partial dislocations slide with two stacking faults along the 1/6 <110> stair-rod dislocation direction, as shown in Figure 7g,h.

## 4. Conclusions

In this research, the MD simulations were carried out to study the response of the interface models composed of 6H-SiC(0001) and four different Al low-index planes to the tensile loading (the four low-index surfaces in this study were (001), (110), (111), and (112)). From the simulation results, it turned out that the changes in the Si or C terminating modes of the SiC(0001) plane had little influence on the tensile properties. However, the interface models respond quite differently to the tensile stress when the four low-index surfaces of Al were used as the variables of the joint surfaces. Because the SiC(0001)/Al(001) models exhibited the highest tensile strength and the smallest elongation, they endured much higher stress under less strain. However, the other models produced certain deformations to relieve the excessive strain, thus increasing the elongation. The SiC(0001)/Al(110) models exhibited the largest elongations among all the models. The combination of interfaces had caused the Al-atoms in the interface region to deviate from their original lattice space, which contributed to the initiation of cracks from the interface region (a specific region two or three atomic layers away from the heterogenous bonding interface in the Al matrix). The SiC(0001)/Al(111) model showed the least lattice mismatch among the four interface models. Thus, the cracks in the SiC(0001)/Al(111) models were more likely to initiate from the Al matrix.

More interesting results were found as further studies were conducted on the four SiC(0001)-Si/Al models. There was almost no plastic deformation detected in the SiC(0001)-Si/Al(001) model. For the SiC(0001)-Si/Al(110) model, simple intersecting stacking faults were formed by the slips of the 1/6 <112> Shockley partial dislocations. Furthermore, dislocation tangles were discovered in the SiC(0001)-Si/Al(111) model due to the dislocation movement. In addition, a unique defect structure composed of three 1/6 <110> stair-rod dislocations and three stacking faults was also discovered during the tensile loading. Lastly, the plastic deformation in the SiC(0001)/Al(112) interface model was restricted by the L-C lock and was carried out along the 1/6 <110> stair-rod dislocations’ direction.

## Figures and Tables

**Figure 1 nanomaterials-13-00404-f001:**
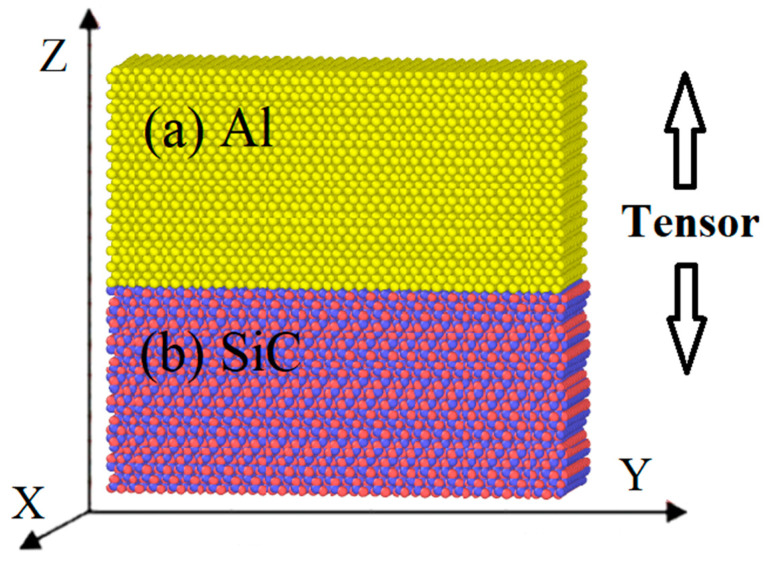
SiC(0001)-Si/Al(001) tensile model structure given as an example: (**a**) Al, (**b**) SiC. Al-atoms are colored yellow, C-atoms are colored red, and Si-atoms are colored blue.

**Figure 2 nanomaterials-13-00404-f002:**
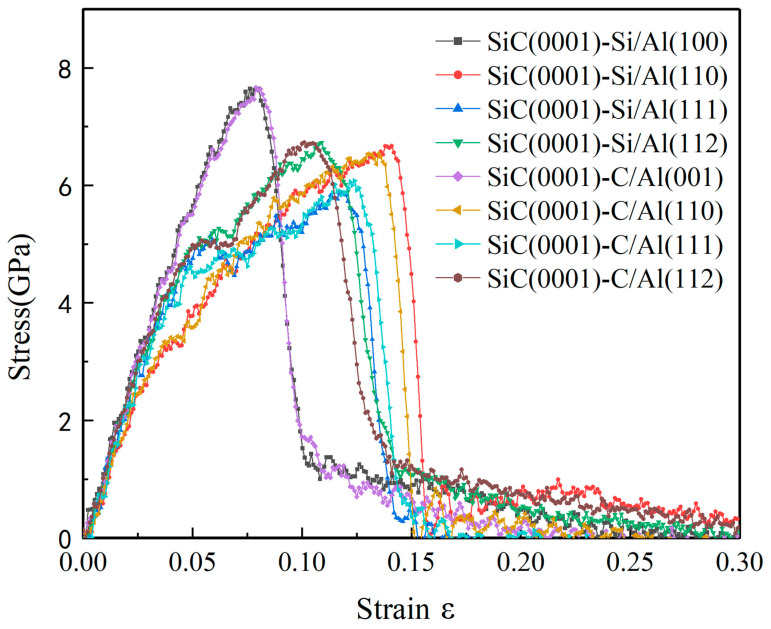
Engineering stress–strain curves of eight interface model systems under tensile loading.

**Figure 3 nanomaterials-13-00404-f003:**
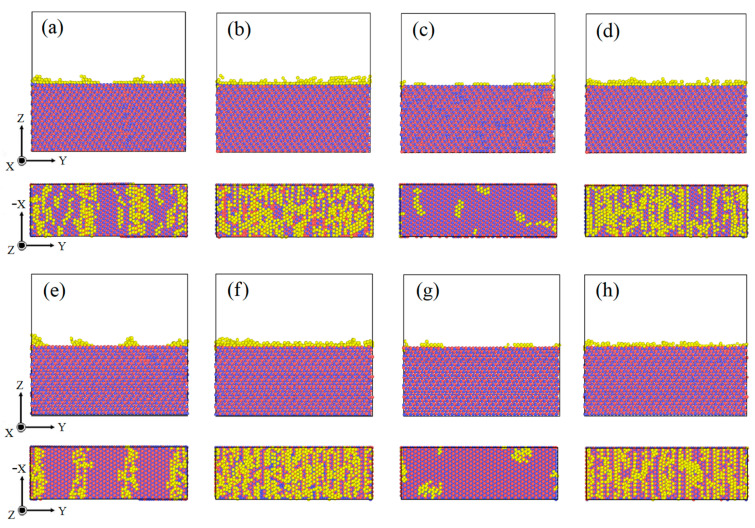
Snapshots of relaxed structures of eight SiC(0001)/Al models colored using the Wigner–Seitz defect analysis. Al-atoms offsetting the original FCC lattice space are colored yellow, C-atoms are colored red, and Si-atoms are colored blue. (**a**) SiC(0001)-Si/Al(001), (**b**) SiC(0001)-Si/Al(110), (**c**) SiC(0001)-Si/Al(111), (**d**) SiC(0001)-Si/Al(112), (**e**) SiC(0001)-C/Al(001), (**f**) SiC(0001)-C/Al(110), (**g**) SiC(0001)-C/Al(111), (**h**) SiC(0001)-C/Al(112).

**Figure 4 nanomaterials-13-00404-f004:**
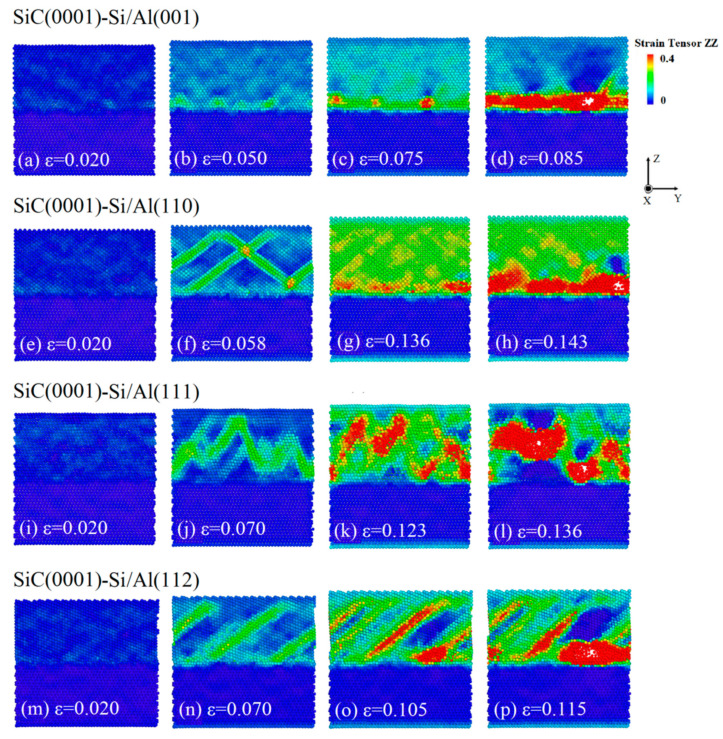
Snapshots of the atomic strain tensor in the Z-direction of four SiC(0001)-Si/Al models at four specific strains: (**a**–**d**) SiC(0001)-Si/Al(001), (**a**) ε = 0.020, (**b**) ε = 0.050, (**c**) ε = 0.075, (**d**) ε = 0.085; (**e**–**h**) SiC(0001)-Si/Al(110), (**e**) ε = 0.020, (**f**) ε = 0.058, (**g**) ε = 0.136, (**h**) ε = 0.143; (**i**–**l**) SiC(0001)-Si/Al(001), (**i**) ε = 0.020, (**j**) ε = 0.070, (**k**) ε = 0.1123, (**l**) ε = 0.136; (**m**–**p**) SiC(0001)-Si/Al(001), (**m**) ε = 0.020, (**n**) ε = 0.070, (**o**) ε = 0.105, (**p**) ε = 0.115.

**Figure 5 nanomaterials-13-00404-f005:**
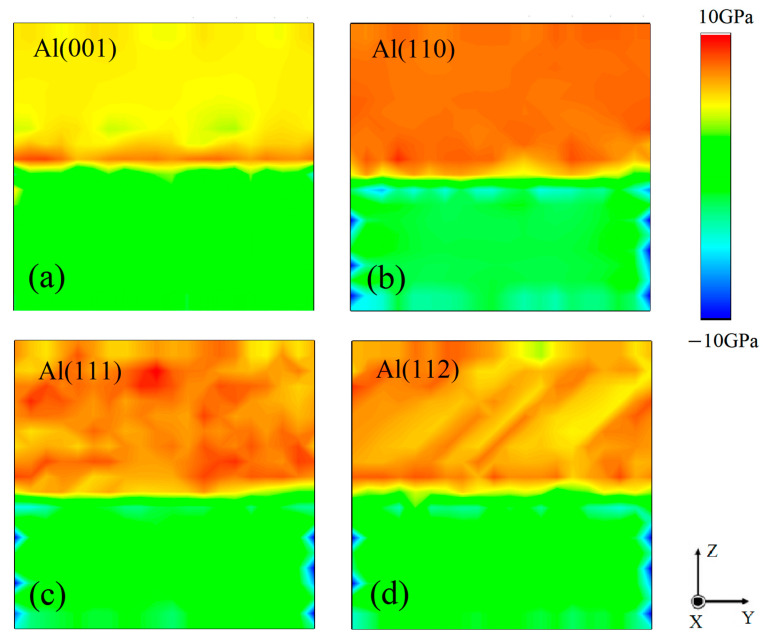
Stress distribution before cracks’ nucleation of four SiC(0001)-Si/Al interface systems: (**a**) SiC(0001)-Si/Al(001), (**b**) SiC(0001)-Si/Al(110), (**c**) SiC(0001)-Si/Al(111), (**d**) SiC(0001)-Si/Al(112).

**Figure 6 nanomaterials-13-00404-f006:**
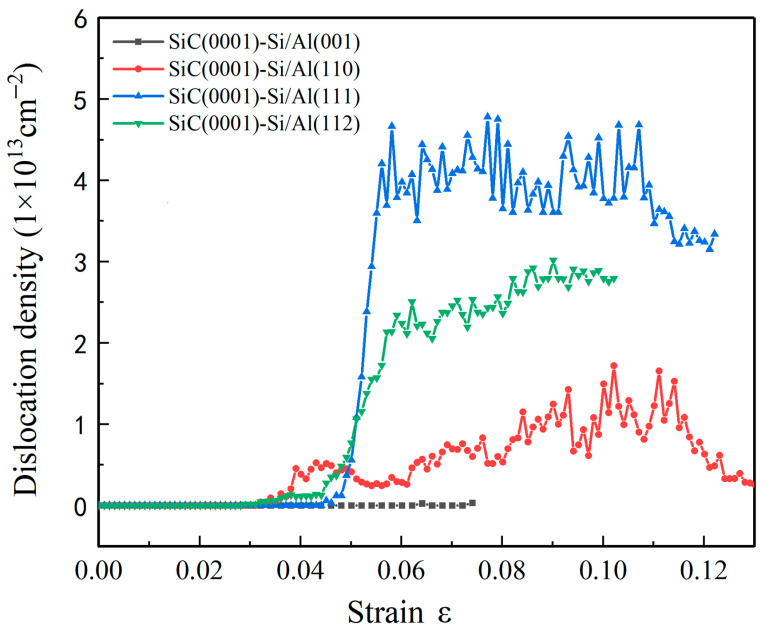
Dislocation density of the four SiC(0001)-Si/Al models during the tensile process.

**Figure 7 nanomaterials-13-00404-f007:**
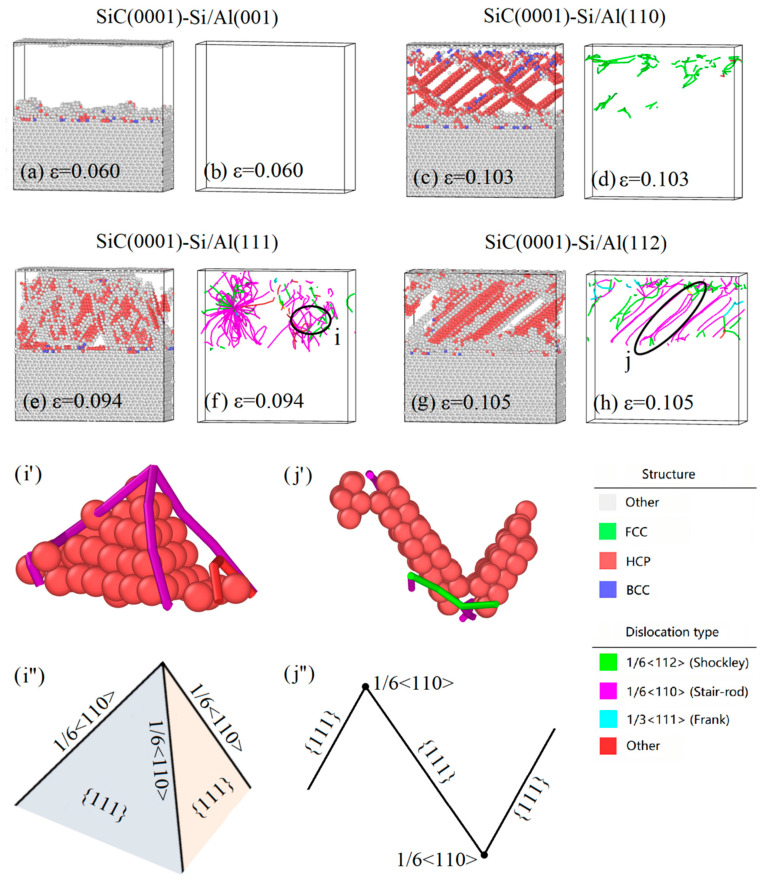
Snapshots of the typical structures and the corresponding dislocation features of (**a**,**b**) SiC(0001)-Si/Al(001) model, (**c**,**d**) SiC(0001)-Si/Al(110) model, (**e**,**f**) SiC(0001)-Si/Al(111) model with (**i**,**i′**,**i″**) defect configuration found in this model, and (**g**,**h**) SiC(0001)-Si/Al(112) model with (**j**,**j′**,**j″**) defect configuration found in this model. The structures are colored using the PTM method. HCP, BCC, and unknown structures are colored red, blue, and white, respectively (with FCC structures neglected). The dislocations are colored using the DXA method. The 1/6 <112> Shockley partial dislocations and 1/6 <110> stair-rod dislocations are colored green and purple, respectively.

**Table 1 nanomaterials-13-00404-t001:** Crystal structures of 6H-SiC and Al.

Element Name	Crystal Structure	Space Group	Lattice Parameters (nm)
Al	FCC	Fm-3m	a = b = c = 0.4049
6H-SiC	Hexagonal	P6_3_mc	a = b = 0.3095; c = 1.5185

**Table 2 nanomaterials-13-00404-t002:** Orientation relationships of the SiC/Al interface systems studied in this work.

6H-SiC	Al	Orientation Relationship	Dimensions X × Y × Z (Å)
(0001)	(001)	${\left(0001\right)}_{SiC}\;\|\;{\left(001\right)}_{Al},\;{\left[11\bar{2}0\right]}_{SiC\;}\|\;{\left[\bar{1}10\right]}_{Al}$	40 × 120 × 110
(0001)	(110)	${\left(0001\right)}_{SiC}\;\|\;{\left(110\right)}_{Al},\;{\left[11\bar{2}0\right]}_{SiC\;}\|\;{\left[001\right]}_{Al}$	40 × 121 × 110
(0001)	(111)	${\left(0001\right)}_{SiC}\;\|\;{\left(111\right)}_{Al},\;{\left[11\bar{2}0\right]}_{SiC\;}\|\;{\left[11\bar{2}\right]}_{Al}$	40 × 120 × 110
(0001)	(112)	${\left(0001\right)}_{SiC}\;\|\;{\left(112\right)}_{Al},\;{\left[11\bar{2}0\right]}_{SiC\;}\|\;{\left[11\bar{1}\right]}_{Al}$	40 × 126 × 110

**Table 3 nanomaterials-13-00404-t003:** Morse potential function parameters parameterized to the ab initio data obtained from Zhao et al. [33].

System	Parameters	Morse Potential
Al-Si	D_o_ (eV)	0.4824
	α (1/Å)	1.322
	r_o_ (Å)	2.92
Al-C	D_o_ (eV)	0.4691
	α (1/Å)	1.738
	r_o_ (Å)	2.246

**Table 4 nanomaterials-13-00404-t004:** Comparisons of the elongations, tensile strength, and Young’s modulus of eight SiC(0001)/Al models.

Si	Al	Elongations	Tensile Strength (GPa)	Young’s Modulus (GPa)
(0001)-Si	(001)	0.080	7.614	119.24
(0001)-Si	(110)	0.141	6.671	91.03
(0001)-Si	(111)	0.120	5.866	103.81
(0001)-Si	(112)	0.109	6.727	112.39
(0001)-C	(001)	0.081	7.641	119.08
(0001)-C	(110)	0.135	6.539	99.15
(0001)-C	(111)	0.124	6.079	101.39
(0001)-C	(112)	0.105	6.670	110.48

## Data Availability

The data will be made available on request.

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
