# Peer review of "Molecular Dynamics Study of Interfacial Micromechanical Behaviors of 6H-SiC/Al Composites under Uniaxial Tensile Deformation"

_nanomaterials, 2023, doi:10.3390/nano13030404_

Round 1

Reviewer 1 Report

The mechanical behaviour of different 6H-SiC/Al systems by means of uniaxial tensile loading has been investigated by using molecular dynamics. Authors found that SiC(00001) plane and Al(111) Plane provides the best match among four interface models, and a unique dislocation structure has been found. For example, a large number of L-C lock structures were found in the case of SiC(00001)/Al(112), which was extended in [110] direction. Authors report that the SiC(00001)/Al(001) provides the highest tensile strength, the smallest elongation, and the highest yield stress. Hence, the overall aim of the study was to compare the different interface models with the tensile stress being a key parameter and identify the best model for these systems.

While the manuscript has scientific merit, a few things require attention before publication:

1) More details about the stress-strain procedure. For example, you can see Physical Chemistry Chemical Physics, 20, 17020 (2018), dot: 10.1039/C8CP03086C

2) It is important to identify the Young modulus from the curves of Fig.2

3) More details on the model, thermostat, etc. for the MD should be provided in the main text or as supplemental information

4) The same applies in the dislocation section. More details are required, for example how dislocations are identified, and what kind of defects their methods are able to detect

5) Authors mention that their force-field is the best and previously validated. May the authors mention a few of the advantages of the chosen force-field in the manuscript?

Author Response

I appreciate very much for your valuable suggestions.

(1) (3) I have added more detailed information about the model, thermostat, and the stress-strain procedure in Chapter 2.1, paragraph 1 and 2, colored in yellow.

(2) See Chapter 3.1, colored in yellow.

(4) See Chapter 2.1, paragraph 3, colored in yellow.

(5) As required by my tutor, I tried reducing the length of the potential part. Tersoff and EAM are very common potential functions for describing SiC and Al atoms respectively [1-3]. As for the morse potential function used to describe the interface, Dandekar et al. [2] has made comparision with the ab initio data and obtained good results.

I improved the manuscript according to your comments. I hope the improvement meet your require. If there is any inappropriate modification, please feel free to inform me.

References

[1] C. Qiu, Y. Su, J. Yang, X. Wang, B. Chen, Q. Ouyang, Di Zhang, Microstructural characteristics and mechanical behavior of SiC (CNT)/Al multiphase interfacial micro-zones via molecular dynamics simulations, Compos B Eng, 220 (2021), Article 108996.

[2] C.R. Dandekar, Y.C. Shin, Molecular dynamics based cohesive zone law for describing Al-SiC interface mechanics, Compos. Part A Appl. Sci. Manuf., 42 (2011), pp. 355-363.

[3] Y. Zhou, W. Yang, M. Hu, Z. Yang, The typical manners of dynamic crack propagation along the metal/ceramics interfaces: a molecular dynamics study, Comput. Mater. Sci., 112 (2016), pp. 27-33.

Reviewer 2 Report

The Authors present an interesting investigation on the interfacial micromechanical behavior of 6H SiC/Al composites under uniaxial tensile deformation. Overall, I think that the research was conducted properly and the results are sound. However, the Authors should clearly state throughout the manuscript the limitations hold by classical force-fields simulations. As an example, the following investigations showed some fundamental discrepancies between classical and ab initio molecular dynamics simulations in determining atomic as well as global forces and behavior in condensed matter. The following investigations may be referenced in the revised manuscript.

1. Physical Chemistry Chemical Physics 21 (15), 8121-8132

2. The Journal of Physical Chemistry Letters 11 (21), 8983-8988

where some limitations of the classical approaches are presented with respect to partially and full quantum molcular dynamics simulations. Once these aspects will be taken into account and the bibliographic apparatus enlarged, it will be my pleasure to re-consider the current manuscript for publication. 

Author Response

I appreciate very much for your valuable suggestions. I improved the manuscript according to your comments. See Chapter 2.2, colored in green.

Reviewer 3 Report

In this article, the tensile properties of 6H-SiC/Al interfaces are studied by performing MD calculations using the LAMMPS software. The authors considered eight models of Si/Al interfaces, SiC(0001) with two terminations and Al(001), (110), (111), and (112), and obtained elongations and tensile strengths. They show that the termination atom type of SiC(0001) has little effect on tensile properties.
On the other hand, tensile properties and deformation behaviors change depending on the Al surface models.

The authors performed a Wigner-Seitz defect analysis and showed that the planes in SiC(0001)/Al(111) model are matched very well. Additionally, the authors discuss in detail the different deformation behaviors of the four models during the tensile processes. Snapshot pictures of atomic strain tensors are shown, and later, atomic structures and dislocations of the models are included to elaborate the discussion. A plot of dislocation density against strain is shown where SiC(0001)-Si/Al(001) shows no noticeable dislocation and moderate to large dislocation density for the other three models.

The MD computational method used in this work is reasonable, and the data provided support their findings. It is scientifically interesting that deformation behaviors are different in these interface models. I have a few suggestions before recommending it for publication.

I recommend authors rewrite the abstract because the finding of this work is all over the place, and it is disorganized. I think the key point of this work is that the tensile properties and deformation behavior vary depending on the interfacing model of the Al surface. The authors can state that first and then write about the highest (or smallest) tensile strength and the elongation of the models and summarize the differences in deformation behaviors of each four models.     

I'd like the author to elaborate on why they've chosen these models to simulate the SiC/Al composites in the introduction section. That motivation is lacking.

I felt that some of the word choices are off. "Elongations are better" should be "elongations are larger." "SiC(0001) plane and Al(111) plane was the best match" may be "SiC(0001)/Al(111) model shows the least lattice mismatch," and so on.

Please define epsilon in Sec. 2. Also, add it in the x-label in Figs. 2 and 6.

In Tables 3 and 4, some values have different significant figures. 

Author Response

I appreciate very much for your valuable suggestions. I improved the manuscript according to your suggestions. I hope the improvement meets your requirements. If there is any inappropriate modification, please feel free to inform me.

1. The abstract has been rewritten.

2. I'd like the author to elaborate on why they've chosen these models to simulate the SiC/Al composites in the introduction section. That motivation is lacking.

For this part, I did not make any modifications. Firstly, it was mentioned in the manuscript that the low index planes are easy to appear in the experiment. Luo [1] had the same opinion and calculated the atomic configuration and coherent energy of Al/SiC interfaces formed of 3C-SiC crystal plane and Al low index plane, (001), (110), and (111). Then Wu [2] studied two common 6H-SiC/Al interfaces by first principles, SiC(0001)/Al(100) and SiC(0001)/Al(111). Romero [3] believes that there is a fixed orientation relationship between the α-SiC and Al, that is, SiC(0001)/Al(112). Considering that there is no paper to compare different interfaces at the microstructure and dislocation aspects, these four Al interfaces are selected. Due to the complexity of the selection process, only their common characteristics are stated in the manuscript. If you insist on adding this part, I will reconsider your recommendation.

[1] X. Luo, G.F. Qian, E.G. Wang, C.F. Chen, Molecular-dynamics simulation of Al/SiC interface structures, Phys. Rev. B, 59 (1999), pp. 10125-10131.

[2] Q.J. Wu, J.P. Xie, A.Q. Wang, D.Q. Ma, C.Q. Wang, First-principle calculations on the structure of 6H-SiC/Al interface, Mater. Res. Express, 6 (2019), Article. 065015.

[3]J.C. Romero et al., Interfacial structure of a SiC/Al composite, Mater Sci Eng A, 1996

3. I felt that some of the word choices are off. "Elongations are better" should be "elongations are larger." "SiC(0001) plane and Al(111) plane was the best match" may be "SiC(0001)/Al(111) model shows the least lattice mismatch," and so on.

See Chapters 3.1, 3.2, and 4.

4. Please define epsilon in Sec. 2. Also, add it in the x-label in Figs. 2 and 6.

See Chapters 2.1 and 3.1, Figs. 2 and 6.

5. In Tables 3 and 4, some values have different significant figures.

This is certainly a place to pay attention to. I have improved my own part in table 4. But the data in Table 3 is not mine, and I do not know its significant figures. I left it what it was because I think it is inappropriate for me to change their data.
